**Data Availability Statement:** All relevant data are within the manuscript.

**Funding:** This study was supported from Scientific Research Cultivation Project of Beijing Municipal

# Evaluation of health-related quality of life and the related factors in a group of Chinese patients with interstitial lung diseases

Xue-Yan Yuan[1][☺], Hui Zhang[1][☺], Li-Ru Huang[1][‡], Fan Zhang[1][‡], Xiao-Wen Sheng[1][‡], Ai Cui[1,2]*

1 Department of Pulmonary and Critical Care Medicine, Beijing Chao-Yang Hospital, Capital Medical University, Beijing, China, 2 Beijing Institute of Respiratory Medicine, Beijing, China

☺ These authors contributed equally to this work.
‡ These authors also contributed equally to this work.
* cuiai@ccmu.edu.cn

## Abstract

### Introduction

Interstitial lung diseases (ILDs) include a wide variety of chronic progressive pulmonary diseases characterized by lung inflammation, fibrosis and hypoxemia and can progress to respiratory failure and even death. ILDs are associated with varying degrees of quality of life impairments in affected people. Studies on the quality of life in patients with ILDs are still limited, and there are few studies with long-term follow-up periods in these patients.

### Methods

Data from patients who were clinically diagnosed with ILDs in the Respiratory Department, Beijing Chaoyang Hospital, Capital Medical University from January 2017 to February 2018 were collected. Clinical status and HRQoL were assessed at baseline and subsequently at 6- and 12-month intervals with the LCQ, mMRC, HADS, SF-36, and SGRQ. Multivariate linear regression was used to evaluate the determinants of the decline in HRQoL.

### Results

A total of 139 patients with idiopathic interstitial pneumonia (IIP) and 30 with connective tissue disease-associated ILD (CTD-ILD) were enrolled, 140 of whom completed the follow-up. The mean age was 63.7 years, and 92 patients were men. At baseline, the decline in HRQoL assessed by the SF-36 and SGRQ was significantly associated with the mMRC, LCQ and HADS depression score. In the follow-up, changes in FVC%, DLco%, mMRC and LCQ were significantly associated with changes in HRQoL.

### Conclusions

HRQoL in both IIP and CTD-ILD patients deteriorates to varying degrees, and the trend suggests that poor HRQoL in these patients is associated with many determinants, primarily

Hospital, Beijing Municipal Administration of Hospitals (No. PX201741) and the funders had no role in study design, data collection and analysis, decision to publish, or preparation of the manuscript.

**Competing interests:** The authors have declared that no competing interests exist.

dyspnea, cough and depression. Improving HRQoL is the main aim when treating patients living with ILDs.

## Introduction

Interstitial lung diseases (ILDs) are a group of chronic and progressive fibrotic lung parenchyma diseases resulting in substantial morbidity and mortality [1, 2]. ILDs include more than 200 subtypes with different etiologies and courses, among which idiopathic interstitial pneumonia (IIP) and connective tissue disease-associated ILD (CTD-ILD) are common subtypes. As the diseases progress, patients' activities of daily living become irreversibly impaired, accompanied by a high symptom burden and significant comorbidities [3]. Meanwhile, the prognosis of these diseases is often poor, which seriously impairs the quality of life (QoL) in affected people due to the insidious onset, lack of typical symptoms, limited therapeutic methods and obvious side effects of medicines [4].

QoL refers to the experience of individuals in different cultures and value systems relating to their goals, expectations, standards and concerns, and it reflects the patient's evaluation of functionality. Health-related quality of life (HRQoL) concerns a person's life satisfaction and happiness as affected by health, including physical, psychological and social functions [5]. Quantifying HRQoL may be helpful in determining patients' subjective understanding of the disease and the disease burden on various aspects of their lives and providing information that cannot be captured by physiologic or radiologic measures.

Previous studies on HRQoL in patients with ILDs focused on idiopathic pulmonary fibrosis (IPF) and sarcoidosis [5–7]. IPF is characterized by an irreversible decline in lung function and death resulting from respiratory failure within 2–3 years [8]. Acute exacerbations of IPF usually have an unidentifiable cause and prodrome, making the clinical course more difficult to predict [9]. As the disease progresses, dyspnea often leads to severe mobility limitations, significantly reducing patients' emotional well-being and independence [5]. Chronic symptoms, poor lung function, and common drug side effects may contribute to the decline in HRQoL [10–12]. Previous studies have shown that HRQoL in these patients deteriorates at various rates [5, 13]. Sarcoidosis is a systemic granulomatous disease with no known cause. As another common subset of ILDs, pulmonary sarcoidosis is associated with a wide range of physical symptoms, such as cough, dyspnea on exertion, and fatigue [14]. Marked deterioration in HRQoL is common in patients with sarcoidosis [7, 15].

Currently, studies about HRQoL in patients with other types of ILDs, such as CTD-ILD and chronic hypersensitivity pneumonia, are rare [16–18]. Few of those conducted were prospective follow-up studies, and most were relatively small cohort studies [17–19]. Research on the HRQoL of Chinese ILD patients is very limited. Additionally, no study has compared the level of HRQoL impairment in IIP and CTD-ILD patients. The aim of this study was to investigate HRQoL in Chinese patients with ILDs including IIP and CTD-ILD and to identify any factors influencing HRQoL among these patients.

## Methods

### Patients and study design

Patients aged ≤85 years who were diagnosed with ILDs including IIP and CTD-ILD at the Department of Respiratory and Critical Medicine of Capital Medical University affiliated with Beijing Chaoyang Hospital from January 2017 to February 2018 were included in the present

prospective study. The diagnosis of IIP or CTD-ILD was based on clinical characteristics and high-resolution computed tomography (HRCT) presentation according to the American Thoracic Society international consensus definition [20]. Patients with the following conditions were excluded: hemorrhagic diseases, New York Heart Association (NYHA) class III to IV heart failure, hepatic insufficiency (alanine aminotransferase level 2 times the upper limit of normal), renal insufficiency (creatinine clearance less than 50 milliliters/minute), and pregnancy or lactation.

A cross-sectional and longitudinal study was conducted, and HRQoL and the factors influencing it were assessed in the patients who were eligible for this study. Then, a prospective cohort study was conducted to follow the changes in the subjects' quality of life, symptoms and physiological indicators every 6 months, and the follow-up lasted for 12 months. Questionnaires were implemented within one week after physiological indicators were measured through face-to face or telephone interviews. All patients were treated according to routine clinical practice, with no additional intervention.

This study was approved by the Institutional Review Board and Ethics Committee of Beijing Chaoyang Hospital, Capital Medical University (2016-Science-149). All patients provided approval and informed consent prior to study entry.

## Clinical data collection

Sociodemographic and disease information was obtained with a standardized questionnaire during clinical examinations at baseline. The following characteristics were included in this questionnaire: the date of birth, age, sex, nationality, height and weight, smoking status, marital status, education, occupation, disease duration, comorbidities, and therapeutic drugs.

Forced expiratory spirometry (forced vital capacity (FVC)), the forced expiratory volume in 1 s ($FEV_1$)) and the diffusing capacity of the lung for carbon monoxide (DLco) were measured according to American Thoracic Society (ATS)/European Respiratory Society (ESR) recommendations (MasterScreen, CareFusion Jaeger, Germany) [21, 22]. The previously established references for FVC, $FEV_1$ and DLco were used [23–25].

Pre-existing recent HRCT images were retrospectively evaluated and scored by two researchers who were blinded to the clinical information [8]. HRCT scoring was originally described by Kazerooni EA et al. and modified according to the actual situation [26]. Briefly, three sections (the section of the aortic arch, the section between the aortic arch and the inferior pulmonary vein, and the section between the inferior pulmonary vein and the diaphragmatic plane) were scored on a scale of 0–5 for ground glass opacities and fibrosis, separately. The percentage was scored as 0 (no finding), 1 ($<5\%$), 2 (5–24%), 3 (25–49%), 4 (50–75%), or 5 ($>75\%$), and interlobular septa thickening was scored as 1 (fibrosis score). The scores for each section were averaged to obtain the final results.

The interstitial lung disease-gender-age-physiology index (ILD-GAP Index) was also assessed from data obtained at the initial evaluation in accordance with the methods proposed by Ryerson et al. [27]. ILD-GAP is an accurate model for predicting mortality in patients with most subtypes and all stages of disease and contains four sets of variables, namely, ILD subtype, sex, age and physiological function. The overall score ranges from 0 to 8; higher scores are associated with higher mortality.

## Questionnaire tests

Dyspnea was measured using the Modified Medical Research Council Dyspnea Scale (mMRC), which has been previously validated. The mMRC is a 5-point scale that asks respondents to rate their dyspnea from 0 (no breathlessness except during strenuous exercise) to 4

(too breathless to leave the house or breathless when dressing or undressing) after receiving an explanation from the staff [28, 29].

Coughing was evaluated with the Chinese version of the Leicester Cough Questionnaire (LCQ), which is a valid instrument for assessing the impact of cough and the ability to detect a response to change. The LCQ is a 19-item self-administered chronic cough QoL questionnaire that includes physical, psychological and social domains, and each represents adverse events caused by cough [30, 31]. It is scored by summing the responses across the three items to form a total score ranging from 3 to 21, with higher scores reflecting less severe cough.

Depression and anxiety were rated using the Hospital Anxiety and Depression Scale (HADS). The HADS is a 14-item questionnaire that contains two subscales with scores on each subscale ranging from 0 to 21 points for anxiety and depression; a score between 8 and 10 indicates borderline caseness, and a score >10 indicates caseness for anxiety and depression [32].

HRQoL was measured using the St. George's Respiratory Questionnaire (SGRQ) and the Short Form-36 (SF-36). The SGRQ is a self-administered, 50-item questionnaire for assessing HRQoL in patients with respiratory diseases; it has previously been used in patients with COPD and IPF [33–35]. It covers three domains: symptoms, activity and impact. The scores for each domain and the total score range from 0 to 100, with higher scores indicating worse quality of life. The SF-36 is a generic questionnaire that contains 36 items categorized into eight domains (vitality, physical functioning, general health, role physical, bodily pain, social functioning, role emotional and mental health) and two psychometrically established summary scores: the physical component score (PCS, constituted by the domains of physical functioning, role physical, general health, and bodily pain) and the mental component score (MCS, constituted by the domains of mental health, emotional role, social functioning, and vitality) [36]. The scores for each domain and summary scores range from 0 to 100, with higher scores indicating better quality of life [37].

## Statistical analysis

In this study, all available data collected at baseline and longitudinally were summarized. Continuous variables are expressed as the mean±standard deviation (SD). Continuous variables with a skewed distribution are expressed as the median and interquartile range (IQR). Categorical variables are expressed as counts and percentages. The characteristics of ILD subtypes (IIP and CTD-ILD) were compared using an unpaired *t* test, a Chi-squared test or the Mann-Whitney's U-test as appropriate. The relationships between the selected variables and baseline HRQoL were characterized by univariate and multivariate linear regression analyses. Variables with a P value <0.10 in the univariate linear regression analysis were included in the multivariate linear model. Multivariate models were constructed using stepwise selection and inverse elimination methods prior to the final assessment of clinical and biological plausibility. In the follow-up, the relationships between changes in HRQoL and clinical characteristics, such as lung function tests and respiratory symptoms, were measured using linear regression analysis. The relationships between changes in the SGRQ total domain and changes in clinical characteristics (stratified into quintiles) were assessed using one-way analysis of variance (ANOVA) followed by pairwise comparisons according to the Least Significant Difference method. A P value <0.05 was considered statistically significant. All statistical analyses were performed with IBM SPSS statistics version 19.

## Results

### Patient characteristics

A total of 169 patients were enrolled in this study, with a median age of 63.7±10.9 years. Sixteen (8%) patients died, and 14 (7%) patients were lost to follow-up; these patients were

excluded from the cohort (Fig 1). Among the included patients, 139 were diagnosed with IIP (101 idiopathic non-specific interstitial pneumonia (NSIP) cases, 18 unclassifiable IIP cases, 11 respiratory-bronchiolitis-ILD cases, 7 IPF cases, and 2 others), and 30 were diagnosed with CTD-ILD (12 Sjögren's syndrome (SS) cases, 6 undifferentiated CTD cases, 4 polymyositis/ dermatomyositis (PM/DM) cases, 3 scleroderma (SSc) cases, 2 rheumatoid arthritis (RA) cases, and 3 others) (S1 Table). No significant sex predominance was noted (male, 54.4%), and most of the included patients (53.8%) were nonsmokers. The duration of symptoms of ILDs was (21.7±28.9) months. Most patients had one (39.6%) or more than one comorbidity. Compared to patients with CTD-ILD, patients with IIP were more likely to be male and to be smokers (Table 1).

The baseline physiological, symptom and psychological characteristics of the patients are presented in Table 2. Compared with patients with IIP, patients with CTD-ILD had more severe lung function impairment as demonstrated by the mean FVC% predicted (mean, 86.9 ±22.2 vs 74.4±19.1; P = 0.017). The mean scores for ground glass opacities and honeycombing were similar in both IIP and CTD-ILD patients. The severity of dyspnea varied greatly in the two groups of patients; 39 patients with IIP were categorized as mMRC 2, closely followed by mMRC 1 (38). Eleven CTD-ILD patients were categorized as mMRC 2. No significant difference in the severity of dyspnea was found between the two groups (P = 0.075). The average severity of cough measured by the LCQ was moderate in both IIP and CTD-ILD patients, with average scores of 16.7±3.7 and 16.3±3.7, respectively. A total of 168 patients completed the evaluation of their psychological problems, and no difference was found between the two groups in the mean HADS-A and HADS-D scores.

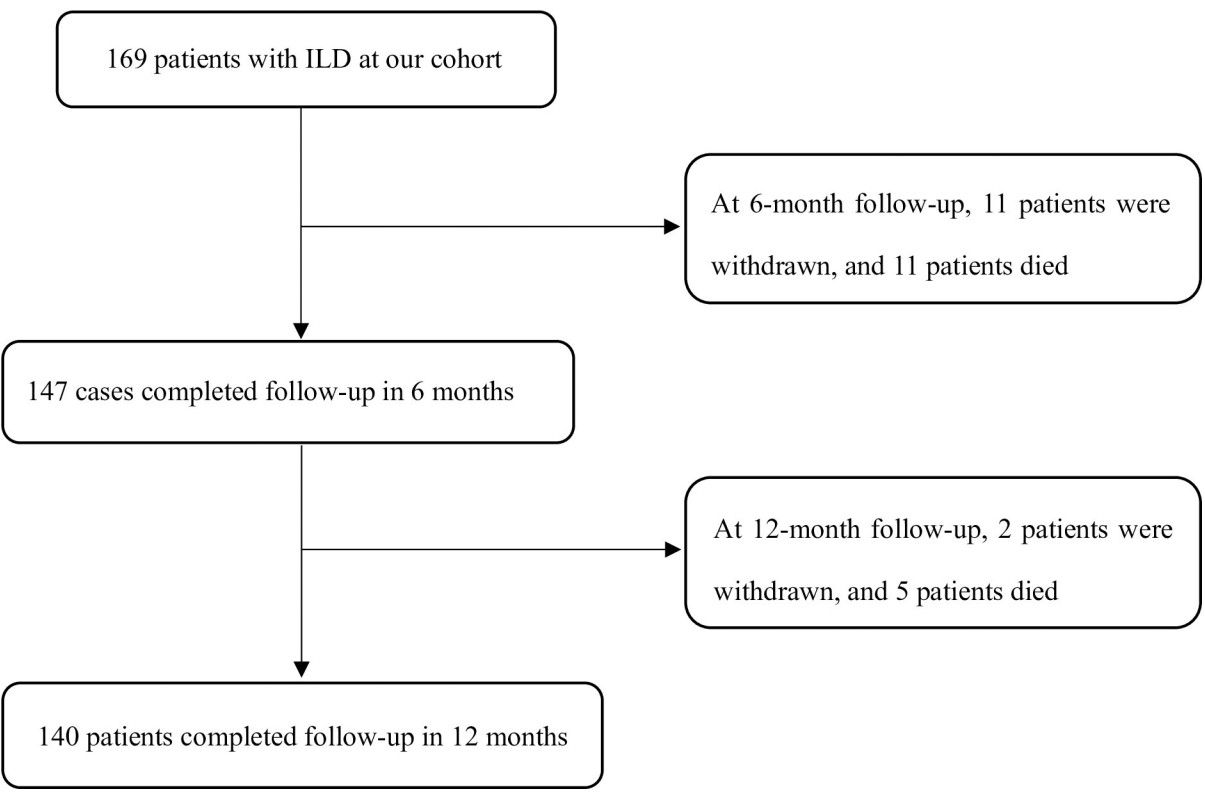

**Fig 1. Flowchart of the study.**

**Table 1. Demographics and disease-related characteristics (n = 169).**

| Characteristics | Value/number (percentage) | | | P value |
|---|---|---|---|---|
| | All patients | IIP | CTD-ILD | |
| N | 169 | 139 | 30 | |
| Age, years | 63.7±10.9 | 64.3±11.0 | 61.1±10.3 | 0.143 |
| Gender, male | 92 (54.4%) | 86 (61.9%) | 6 (20.0%) | 0.000 |
| Ethnic group | | | | 0.203 |
| Han | 156 (92.3%) | 130 (93.5%) | 26 (86.7%) | |
| Other | 13 (7.7%) | 9 (6.5%) | 4 (13.3%) | |
| BMI, kg/m$^2$ | 25.2±4.1 | 25.4±4.0 | 24.4±4.3 | 0.227 |
| Education level* | | | | 0.203 |
| Low | 41 (24.3%) | 31 (22.3%) | 10 (33.3%) | |
| Middle-High | 128 (75.7%) | 108 (77.7%) | 20 (66.7%) | |
| Economic situation, RMB/year/family | | | | 0.026 |
| <5000 | 99 (58.6%) | 77 (55.4%) | 22 (73.3%) | |
| 5000–10000 | 53 (31.4%) | 46 (33.1%) | 7 (23.3%) | |
| >10000 | 17 (10.1%) | 16 (11.5%) | 1 (3.3%) | |
| Smoking history | | | | 0.003 |
| Current | 22 (13.0%) | 22 (15.8%) | 0 (0.0%) | |
| Former | 56 (33.1%) | 50 (36.0%) | 6 (20.0%) | |
| Never | 91 (53.8%) | 67 (48.2%) | 24 (80.0%) | |
| Disease duration, months | 21.7±28.9 | 20.7±26.9 | 26.3±37.3 | 0.442 |
| Number of comorbidities** | | | | 0.218 |
| 0 | 56 (33.1%) | 42 (30.2%) | 14 (46.7%) | |
| 1 | 67 (39.6%) | 58(41.7%) | 9 (30.0%) | |
| ≥2 | 46 (27.2%) | 39 (28.1%) | 7 (23.3%) | |
| Therapeutic drugs used previously | | | | 0.072 |
| Corticosteroids or/and immunosuppressants | 34 (20.1%) | 24 (17.3%) | 10 (33.3%) | |
| Antifibrotic drugs | 5 (3.0%) | 4 (2.9%) | 1 (3.3%) | |
| No intervention | 113 (66.9%) | 99 (71.2%) | 14 (46.7%) | |
| Others*** | 17 (10.1%) | 12 (8.6%) | 5 (16.7%) | |

Data are expressed as a number (%) or the mean±SD. BMI, body mass index; IIP, idiopathic interstitial pneumonia; CTD, connective tissue disease; CTD-ILD, CTD-associated ILD; HP, hypersensitivity pneumonitis; COPD, chronic obstructive pulmonary disease

* A low education level indicates that patients received only primary education, while a middle-high education level indicates that patients received secondary education or above.

** Comorbidities included asthma, pulmonary hypertension, COPD/emphysema, lung cancer, pulmonary embolism, gastro-esophageal reflux disease, cardiovascular disease and metabolic diseases.

*** Other drugs included traditional Chinese medicine and antioxidants

## HRQoL

The decline in HRQoL was significant in most dimensions in both IIP and CTD-ILD patients (Table 2). Regarding the SF-36, patients with CTD-ILD had more impaired HRQoL than patients with IIP as assessed by the SF-36 PCS (mean, 31.1±14.2 vs 37.2±12.0; P = 0.015). HRQoL, as measured by the SF-36 MCS, was similar in the two groups. At baseline, patients with CTD-ILD had higher scores in all SGRQ dimensions except for the symptom domain (mean, 40.7±25.8 vs 37.4±23.9; P = 0.494). The different dimensions of HRQoL measured with the SF-36 at baseline for all patients are presented in Fig 2. Compared to the other three dimensions in the PCS, the mean score was the lowest for general

**Table 2. Clinical data of the ILD patients at the time of enrollment.**

| Variable | Value | | P value |
|---|---|---|---|
| | IIP | CTD-ILD | |
| HRQoL Questionnaire | | | |
| mMRC (0/1/2/3/4) (n = 169) | 37/38/39/16/9 | 4/7/11/8//0 | 0.075 |
| LCQ domain (n = 168) | | | |
| Physical domain | 5.4±1.3 | 5.3±1.3 | 0.555 |
| Psychological domain | 5.5±1.3 | 5.4±1.3 | 0.574 |
| Social domain | 5.8±1.3 | 5.7±1.3 | 0.699 |
| Total domain | 16.7±3.7 | 16.3±3.7 | 0.566 |
| SF-36 (n = 169) | | | |
| Physical functioning | 80.0[60.0, 90.0] | 65.0[40.0, 80.0] | 0.010 |
| Role physical | 25.0[0.0, 81.2] | 50.0[0.0, 100.0] | 0.158 |
| Bodily pain | 72.0[40.0, 100.0] | 56.5[40.0, 88.0] | 0.419 |
| General health | 45.0[30.0, 55.0] | 30.0[15.0, 57.0] | 0.049 |
| Vitality | 60.0[45.0, 75.0] | 55.0[40.0, 60.0] | 0.023 |
| Social functioning | 75.0[62.5, 100.0] | 68.7[50.0, 90.6] | 0.216 |
| Role emotional | 66.7[0.0, 100.0] | 100.0[33.3, 100.0] | 0.134 |
| Mental health | 64.0[53.0, 72.0] | 68.0[52.0, 80.0] | 0.129 |
| PCS | 37.2±12.0 | 31.1±14.2 | 0.015 |
| MCS | 48.3±11.6 | 45.6±11.1 | 0.231 |
| SGRQ (n = 169) | | | |
| Symptom | 37.4±23.9 | 40.7±25.8 | 0.494 |
| Activity | 40.0±26.0 | 54.2±29.0 | 0.009 |
| Impact | 26.8±19.2 | 37.2±20.3 | 0.008 |
| Total | 32.9±19.1 | 43.3±20.6 | 0.009 |
| Symptom score | | | |
| Pulmonary function (n = 139) | | | |
| FVC, % predicted | 86.9±22.2 | 74.4±19.1 | 0.017 |
| $FEV_1$/FVC | 82.3±9.7 | 81.3±6.8 | 0.658 |
| DLco, % predicted | 60.9±18.1 | 52.9±13.6 | 0.058 |
| ILD-GAP index (n = 135) | 2.0[1.0, 3.2] | 0.0[0.0, 1.0] | 0.000 |
| Chest CT image (n = 168) | | | |
| Ground glass opacity score | 2.7±1.1 | 2.9±0.9 | 0.304 |
| Honeycombing score | 1.4±1.1 | 1.4±0.8 | 0.953 |
| HADS-A (n = 168) | 5.0[3.0, 7.0] | 6.0[3.0, 9.0] | 0.245 |
| HADS-D (n = 168) | 5.0[1.0, 7.0] | 5.5[2.7, 9.2] | 0.086 |

Data are expressed as a number, the mean±SD, or the median (interquartile range). ILD, interstitial lung disease; $FEV_1$, forced expiratory volume in 1 s; FVC, forced vital capacity; DLco, diffusing capacity of the lung for carbon monoxide; ILD, interstitial lung disease; ILD-GAP, The interstitial lung disease-gender-age-physiology index; HRQoL, Health-related Quality of Life; mMRC, modified Medical Research Council dyspnea scale; LCQ, Leicester Cough Questionnaire; HADS, Hospital Anxiety and Depression Scale; HADS-A, HADS-Anxiety; HADS-D, HADS-Depression; SF-36, the Medical Outcomes Study Short Form 36; PCS, physical component score; MCS, mental component score; SGRQ, St. George's Respiratory Questionnaire

health. On average, the score was lowest in the vitality domain and highest in the social functioning domain on the MCS.

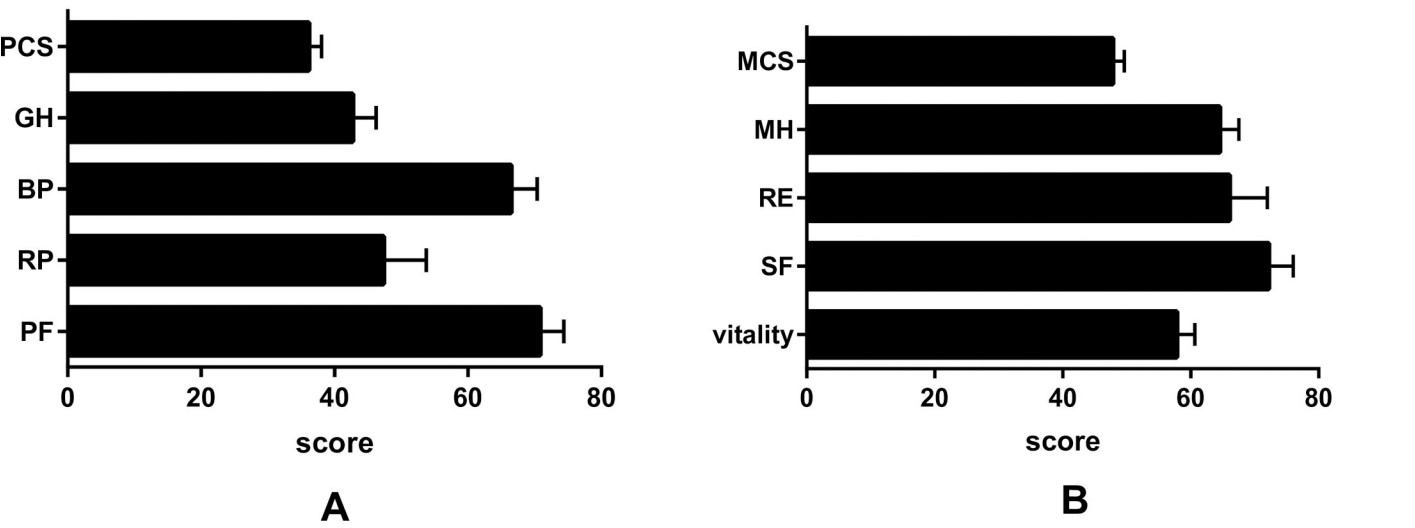

**Fig 2. HRQoL scores at baseline by the SF-36 for all patients (n = 169).**

A. The mean scores for dimensions related to physical health; B. The mean scores for dimensions related to mental health (SF-36, the Medical Outcomes Study Short Form 36; PCS, physical component score; MCS, mental component score).

### Factors influencing HRQoL assessed at the time of enrollment

As shown in Table 3, HRQoL was found to be significantly affected by multiple factors. ILD subtype was negatively associated with the SF-36 PCS and positively associated with most dimensions of the SGRQ in the univariate liner regression analyses. As previously mentioned, patients with CTD-ILD had a lower quality of life as measured by the SF-36 PCS and SGRQ activity, impact, and total domains when compared with patients with IIP. Sociodemographic factors, such as sex, age, education level and smoking history, were found to be associated to some degree with part of the dimensions of HRQoL calculated by the SGRQ and SF-36. Disease durations and therapeutic drugs were associated with most dimensions of HRQoL according to the SGRQ results. The FVC% predicted and $DL_{CO}$% predicted were strongly associated with most dimensions of the SGRQ except for the symptom domain and the SF-36 PCS. Ground glass opacity on chest CT was associated with one dimension of HRQoL (the SF-36 PCS), and honeycombing was independently associated with most dimensions of HRQoL (SGRQ). Typical symptoms of disease, including dyspnea and cough, were strongly associated with most dimensions of HRQoL (SGRQ and SF-36 PCS). Psychological factors were demonstrated to be associated with some dimensions of HRQoL (SGRQ and SF-36). The ILD-GAP score was also found to be associated with all dimensions of the SGRQ and the SF-36 PCS.

Multivariate linear regression analysis showed that several dimensions of the SGRQ were significantly associated with the mMRC score, LCQ total domain, and HADS-D, whereas dimensions of the SGRQ were weakly associated with clinical variables including sex, BMI, disease duration, and the DLco% predicted. After adjustment for mMRC, ILD subtype (IIP vs CTD-ILD) remained independently associated with SF-36 PCS (P≤0.05). A significant association between the SF-36 PCS and mMRC score was identified. The SF-36 MCS was significantly associated with both the HADS-A and HADS-D (Table 4).

**Table 3. Association between HRQoL and other measures at baseline: Results of the univariate linear regression analysis (n = 169).**

| Characteristics | SGRQ | | | | SF-36 | |
|---|---|---|---|---|---|---|
| | Total | Symptom | Activity | Impact | PCS | MCS |
| ILD subtypes | 10.302** | 3.322 | 14.023** | 10.342** | -6.091* | -2.868 |
| Sex | 4.710 | -1.941 | 9.844* | 3.606 | -5.597** | -0.334 |
| Age, years | 0.231 | 0.067 | 0.389* | 0.185 | -0.126 | 0.033 |
| BMI, kg/m$^2$ | -0.317 | 0.870 | -0.585 | -0.522 | 0.392 | 0.302 |
| Education level | -13.05*** | -12.09** | -14.39** | -12.67*** | 7.228*** | -0.995 |
| Economic level | -2.860 | -2.422 | -3.300 | -2.823 | 1.428 | 0.455 |
| Smoking history | 2.571 | -2.675 | 4.757 | 2.903 | -2.795* | -1.631 |
| Disease duration, months | 0.141** | 0.152* | 0.197** | 0.103 | -0.079* | 0.051 |
| Therapeutic drug | -3.383* | -4.406* | -1.265 | -4.358** | -0.152 | 0.417 |
| Pulmonary function | | | | | | |
| FVC, % predicted | -0.346*** | -0.133 | -0.497*** | -0.323*** | 0.195*** | 0.051 |
| FEV$_1$/FVC | -0.057 | -0.423 | 0.112 | -0.056 | -0.133 | -0.103 |
| DLco, % predicted | -0.546*** | -0.278* | -0.812*** | -0.469*** | 0.313*** | 0.006 |
| Chest CT image | | | | | | |
| Ground glass opacity | 1.043 | 1.216 | 3.038 | -0.187 | -1.869* | -0.030 |
| Honeycombing | 3.285* | 3.632* | 3.014 | 3.401* | -0.945 | 0.691 |
| Number of comorbidities | 0.488 | 1.913 | -0.040 | 0.361 | -0.133 | -0.028 |
| mMRC | 13.224*** | 7.890*** | 19.974*** | 10.762*** | -8.347*** | -0.616 |
| LCQ total domain | -3.549*** | -3.053*** | -3.580*** | -3.694*** | 1.340*** | 0.554* |
| HADS | | | | | | |
| HADS-A | 1.863*** | 1.319* | 1.609** | 2.170*** | -0.339 | -1.483*** |
| HADS-D | 2.156*** | 0.791 | 2.712*** | 2.228*** | -0.897*** | -1.336*** |
| ILD-GAP | 4.398*** | 2.962* | 5.208*** | 4.317*** | -4.299*** | -0.023 |

Data are presented as the beta estimates of regression coefficients. BMI, body mass index; FEV$_1$, forced expiratory volume in 1 s; FVC, forced vital capacity; DLco, diffusing capacity of the lung for carbon monoxide; mMRC, modified Medical Research Council dyspnea scale; LCQ, Leicester Cough Questionnaire; HADS, Hospital Anxiety and Depression Scale; HADS-A, HADS-Anxiety; HADS-D, HADS-Depression; ILD-GAP, The interstitial lung disease-gender-age-physiology index.

* P≤0.05,

** P≤0.01,

*** P≤0.001

## Relationship between the change in HRQoL and changes in clinical characteristics

Changes in pulmonary function, the dyspnea score and the cough score were assessed at 6 months and 12 months (S2 Table). At 6 months, 24 patients' FVC% predicted and 35 patients' DLco% predicted were stable. A total of 31 patients had improvements in dyspnea, and 62 patients had experienced relief from coughing. Similarly, 14 patients' FVC% predicted and 23 patients' DLco% predicted were stable at the 12-month follow-up. A total of 32 patients experienced relief from dyspnea, and 54 patients experienced relief from coughing at the 12-month follow-up.

At the 6-month follow-up, the changes in HRQoL measured by the SF-36 MCS and SGRQ impact domains revealed a mild improvement in quality of life among patients with IIP. Patients with CTD-ILD had mildly improved HRQoL, as measured by all SF-36 and SGRQ domains except for the symptom domain. Furthermore, IIP patients had mildly improved quality of life as measured by the SF-36 MCS and SGRQ impact domains at the 12-month follow-up. The change in HRQoL measured by the SGRQ activity domain demonstrated a mild

**Table 4. Associations between HRQoL and other measures at baseline: Results of the multivariate linear regression analysis (n = 169).**

| | SGRQ | | | | SF-36 | |
|---|---|---|---|---|---|---|
| | **Total** | **Symptom** | **Activity** | **Impact** | **PCS** | **MCS** |
| $R^2$ | 0.727 | 0.392 | 0.770 | 0.617 | 0.528 | 0.269 |
| ILD subtypes | | | | | -2.300* | |
| Sex | | | 6.584** | | | |
| BMI | | 0.962* | | | | |
| Disease duration | | 0.153* | | | | |
| DLco, % predicted | | | -0.171* | | | |
| mMRC | 9.098*** | 4.826** | 16.131*** | 4.868*** | -0.694*** | |
| LCQ total domain | -2.233*** | -2.725*** | -0.976** | -2.825*** | | |
| HADS-A | | | | | | -0.230** |
| HADS-D | 0.690** | | 0.768** | 0.897*** | | -0.351*** |

Data are presented as the beta estimates of regression coefficients. BMI, body mass index; DLco, diffusing capacity of the lung for carbon monoxide; mMRC, modified Medical Research Council dyspnea scale; LCQ, Leicester Cough Questionnaire; HADS, Hospital Anxiety and Depression Scale; HADS-A, HADS-Anxiety; HADS-D, HADS-Depression; ILD-GAP, The interstitial lung disease-gender-age-physiology index.

* $P \leq 0.05$,

** $P \leq 0.01$,

*** $P \leq 0.001$

decrease in IIP patients. Changes in HRQoL among CTD-ILD patients revealed a mild to moderate improvement in quality of life as measured by all SF-36 domains and the SGRQ total and impact domains (Fig 3).

The associations between longitudinal changes in HRQoL and clinical characteristics, including the FVC% predicted, DLco% predicted, mMRC and LCQ total score, are shown in Table 5. At 6 months, the change in clinical characteristics was associated with changes in the SGRQ

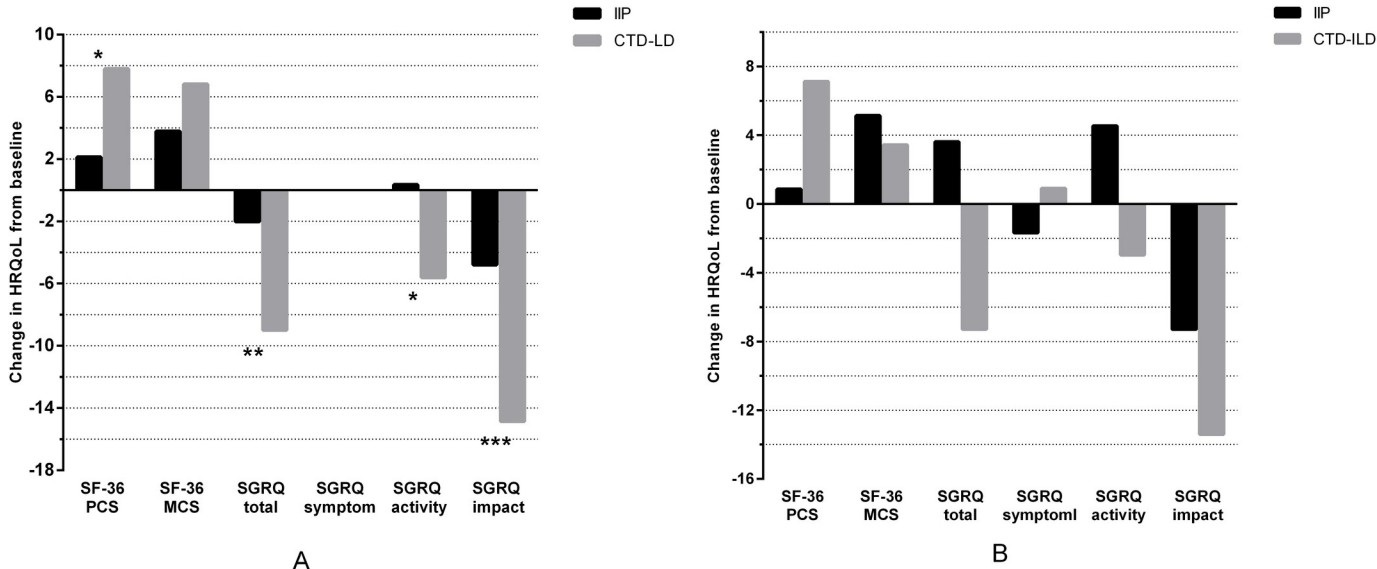

**Fig 3. HRQoL scores measured by the SF-36 and SGRQ at the 6-month follow-up (n = 147) and the 12-month follow-up (n = 140).** A. Changes in HRQoL from baseline to the 6-month follow-up; B. Changes in HRQoL from baseline to the 12-month follow-up. (SF-36, the Medical Outcomes Study Short Form 36; PCS, physical component score; MCS, mental component score; SGRQ, St. George's Respiratory Questionnaire. * $P \leq 0.05$, ** $P \leq 0.01$, *** $P \leq 0.001$).

**Table 5. Association between the change in HRQoL and clinical characteristics at the 6-month follow-up and the 12-month follow-up.**

| | ΔSGRQ total | | ΔSGRQ symptom | | ΔSGRQ activity | | ΔSGRQ impact | |
|---|---|---|---|---|---|---|---|---|
| | 6-month follow-up | 12-month follow-up | 6-month follow-up | 12-month follow-up | 6-month follow-up | 12-month follow-up | 6-month follow-up | 12-month follow-up |
| ΔFVC% predicted | -0.481*** | -0.412*** | -0.059 | -0.223* | -0.551*** | -0.484*** | -0.377*** | -0.318** |
| ΔDLco% predicted | -0.458*** | -0.554*** | 0.013 | -0.253* | -0.559*** | -0.553*** | -0.355** | -0.514*** |
| ΔmMRC | 0.641*** | 0.706*** | 0.082 | 0.399*** | 0.692*** | 0.755*** | 0.537*** | 0.578*** |
| ΔLCQ | -0.562*** | -0.584*** | -0.239** | -0.420*** | -0.404*** | -0.411*** | -0.559*** | -0.595*** |

Data are presented as the beta estimates of regression coefficients. At the 6-month follow-up, 81 patients completed the pulmonary function assessment, although ΔDLco% predicted data were not available for 4 patients. A total of 147 patients completed the LCQ assessment, and 146 patients completed the mMRC assessment. All data were available. At the 12-month follow-up, 79 patients completed the pulmonary function assessment, although ΔDLco% predicted data were not available for 6 patients. A total of 140 patients completed the mMRC and LCQ assessments, and all data were available. FVC, forced vital capacity; ΔFVC% predicted, FVC change from baseline at 6-month follow-up; DLco, diffusing capacity of the lung for carbon monoxide; ΔDLco % predicted, DLco change from baseline at the 6-month follow-up; mMRC, modified Medical Research Council dyspnea scale; ΔmMRC, mMRC change from baseline at the 6-month follow-up; LCQ, Leicester Cough Questionnaire; ΔLCQ, LCQ change from baseline at the 6-month follow-up.

* P≤0.05,

** P≤0.01,

*** P≤0.001

activity domain, impact domain, and total domain. A relationship between changes in clinical characteristics (the FVC% predicted, DLco% predicted, mMRC) and changes in SGRQ symptoms was not found. At the 12-month follow-up, the moderately significant relationship between changes in clinical characteristics and changes in all dimensions of the SGRQ were confirmed. Changes in pulmonary function (FVC% predicted, DLco% predicted) and the LCQ total score were negatively associated with changes in all dimensions of the SGRQ, while changes in the mMRC score were positively associated with changes in all dimensions of the SGRQ.

### Clinical characteristics and HRQoL of patients who died during follow-up

Of the 169 included patients, 16 patients died during the follow-up (S3 Table). The age at death was 66.7±8.3 years, 10 patients (62.5%) were male, and the duration from the onset of the first symptom to death was 27.8±21.0 months. The diagnosis of most patients who died (15, 93.8%) was IIP, and the main cause of death (10, 62.5%) was pulmonary infection, followed by AE-ILD (5, 31.25%). Differences in clinical characteristics including age, sex, and duration since the first symptoms were not significant between the survivors and non-survivors. In addition, the ILD subtype was not associated with the prognosis in the study population. The HRQoL in patients who died was significantly worse than that in survivors as measured by the SF-36 and SGRQ. The mean SF-36 PCS score in non-survivors was significantly lower than that in survivors (27.1 vs 38.1, P = 0.000). The mean SGRQ total score was significantly higher in non-survivors than in survivors (53.8 vs 28.5, P = 0.000).

### Discussion

ILDs include more than 200 subtypes with different prognoses, which not only significantly shorten the survival time but also impair quality of life in patients [4, 19, 37]. According to the present data, most aspects of HRQoL in patients with IIP and CTD-ILD, as measured by the SF-36 and SGRQ, were moderately to severely reduced, and impairment of HRQoL was even more pronounced in patients with CTD-ILD compared with patients with IIP. Physical aspects

measured by the SF-36 PCS and SGRQ activity domains were the most impaired in all included patients. Meanwhile, the results of the comprehensive data analysis suggested that the cause of the decline in HRQoL in patients with ILDs was complex and multifactorial. Close associations between HRQoL and symptom severity and psychological deficits was found. In addition, other factors, including ILD subtypes (IIP and CTD-ILD), sex, BMI, disease duration and the DLco% predicted, had mild to moderate associations with HRQoL in these patients. In our study, the data further showed that changes in HRQoL were significantly associated with changes in pulmonary function and symptoms, including predicted FVC%, predicted DLco%, dyspnea and cough.

ILDs represent a heterogeneous group of conditions characterized by varying degrees of inflammation and pulmonary fibrosis. They may either appear as an idiopathic condition termed IIP or in association with CTD. The present study supported previous studies that showed IIP and CTD-ILD are clinically similar, with insidious onset of dyspnea as the main clinical manifestation [38, 39]. ILDs remain difficult to treat and are associated with reduced quality of life and mortality [40]. As shown in the present study, the quality of life decreased significantly in all patients, although impairment was more severe in patients with CTD-ILD than patients with IIP.

Similar to our finding that the mean SGRQ total score in IIP patients was 32.9 at baseline, a previous study conducted by Furukawa et al. [41] reported that the mean SGRQ total score in patients with IPF was 34.5. Michael et al. [42] reported that the mean SGRQ total score in 623 IPF patients (48.3) was significantly higher than our result, which indicated a worse quality of life. This could be partly explained by the differences in age, race, duration of disease and ILD subtypes in the cohorts. Patients with CTD-ILD often experience impaired HRQoL. In a previously published study, the quality of life of 193 patients with CTD-ILD was significantly decreased with a SGRQ total score of 36.3. In one study of 177 patients with SSc, the SGRQ total score for the SSc-ILD subgroup was 30.2, which is lower than the present result (43.3) [43]. A comparison of quality of life between IIP and CTD-ILD patients was not available before the present study, and further exploration is needed.

Dyspnea and cough are common symptoms in ILD patients, and previously published studies have indicated that cough, dyspnea and depression are potentially associated with HRQoL in ILD patients [44–46]. In the study involving the Australian IPF Registry, Glaspole et al. [11] compiled and analyzed the data from 516 patients and found that dyspnea, cough and depression were three major contributors to HRQoL. Multivariate analysis of our study data corroborated previous results, finding that dyspnea, cough and depression were the major determinants. Furthermore, mild to moderate associations between HRQoL and other measures, including ILD subtypes, sex, BMI, disease duration and the DLco% predicted, were demonstrated in our study, unlike in earlier studies. To the best of our knowledge, ventilatory impairment suggested by poor pulmonary function in ILD may further impair quality of life [4]. However, there is no consensus on the relationship between HRQoL and pulmonary function. Based on multivariate analysis of HRQoL at baseline, a mild association was identified between HRQoL and the DLco% predicted in our study, which is inconsistent with the study result from the insights-IPF registry, which showed moderately strong associations between HRQoL and the FVC% predicted and DLco% predicted [42]. Similar to our results, the results from the INPULSIS trials demonstrated that HRQoL measured by the SGRQ was weakly associated with FVC% predicted at baseline [47].

Although earlier studies had demonstrated a decline in HRQoL in ILD patients and multifactorial interactions between HRQoL and clinical characteristics, few of them explored changes in HRQoL in ILD patients during long-term follow-up [47, 48]. In our cohort, longitudinal data on HRQoL assessed by the SF-36 and SGRQ revealed that HRQoL had weakly

improved from baseline at both the 6-month follow-up and the 12-month follow-up in contrast to the results of a recently completed longitudinal study conducted by Rajala et al. [49]. The difference may be explained in part by the different subtypes, phases of diseases and pharmacotherapy. Despite the lack of an association between HRQoL and pulmonary function at baseline, our results demonstrated that changes in HRQoL measured by the SGRQ had a significant association with changes in pulmonary function, and the associations among SGRQ activity, impact, and total scores and FVC% predicted and DLco% predicted were statistically strong. Our result is consistent with the results from the INPULSIS trial and the insights-IPF registry [10, 43], both of which showed that the change in HRQoL assessed by the SGRQ total score was significantly associated with a decline in the FVC% predicted of >10%. In addition, the decline in HRQoL was significant in patients who experienced a decline in the FVC% predicted that was >5% in our analysis.

In our cohort, the study found that patients who died during follow-up had a worse HRQoL regardless of the subtype and phase of disease at baseline compared to patients who survived. There have been few studies on HRQoL and survival in patients with ILDs. A previous study including 182 IPF patients demonstrated that HRQoL assessed by the SGRQ total score and the FVC% predicted at baseline were independent prognostic predictors of mortality (HR, 1.012; P = 0.029) [44]. In the Finnish IPF study, 37% of the 247 included patients died during follow-up, and in those patients, HRQoL as measured by the RAND-36 deteriorated significantly in all dimensions except physical role [49].

There were several notable limitations in our study that need to be addressed. First, our study did not include the 6-minute walking distance (6MWD) and the NYHA functional status, which are the tools used to assess functional exercise capacity. Previous studies have demonstrated that both missing indicators are clinically meaningful predictors of HRQoL in patients with ILDs. The absence of these indicators may affect our analysis to some extent. Second, some patients did not complete each of the HRQoL questionnaires during follow-up. Additionally, pulmonary function data were incomplete during follow-up. All of these factors may lead to skewing of the analysis. Third, the patient-reported outcome measures (PROs) used in this study (the mMRC, LCQ, HADS, SGRQ and SF-36) were not originally developed for IPF and ILDs, and the minimal clinically significant differences of the PROs are currently unknown, suggesting that further studies are needed to confirm the validity of the PROs in IPF and ILDs. Fourth, the lack of therapeutic factors during the follow-up in our study may be responsible for the improved quality of life in some patients at the follow-up visits. Finally, although the finding that the decline in HRQoL was significantly associated with clinical symptoms, depression and changes in pulmonary function, further study regarding whether management of these determinants could improve HRQoL was not performed.

## Conclusions

In conclusion, our findings show that HRQoL in patients with IIP and CTD-ILD deteriorates at various rates, and the decline of patients with CTD-ILD was more significant. Moreover, our findings demonstrate that the determinants of the decline in HRQoL are multifactorial; the major determinants are dyspnea, cough and depression. Furthermore, changes in HRQoL are significantly associated with changes in pulmonary function. Improving HRQoL is the main aim when treating patients living with ILDs.

## Supporting information

**S1 Table. IIP and CTD-ILD subgroups.**
(DOCX)

**S2 Table. Associations between changes in HRQoL and clinical characteristics.**
(DOCX)

**S3 Table. Comparison of the main demographics, clinical characteristics and HRQoL according to the prognosis.**
(DOCX)

## Acknowledgments

We are thankful for the efforts of all the participants and physicians who contributed to this study, as well as the patients who participated in this study.

## Author Contributions

**Conceptualization:** Ai Cui.

**Data curation:** Xue-Yan Yuan, Hui Zhang, Li-Ru Huang, Fan Zhang, Xiao-Wen Sheng.

**Formal analysis:** Xue-Yan Yuan, Hui Zhang.

**Funding acquisition:** Ai Cui.

**Investigation:** Xue-Yan Yuan, Ai Cui.

**Methodology:** Xue-Yan Yuan, Hui Zhang, Ai Cui.

**Project administration:** Xue-Yan Yuan, Hui Zhang, Ai Cui.

**Resources:** Xue-Yan Yuan, Hui Zhang, Ai Cui.

**Software:** Li-Ru Huang, Fan Zhang, Xiao-Wen Sheng.

**Supervision:** Xue-Yan Yuan, Ai Cui.

**Validation:** Xue-Yan Yuan, Hui Zhang.

**Visualization:** Xue-Yan Yuan, Li-Ru Huang.

**Writing – original draft:** Xue-Yan Yuan, Hui Zhang.

**Writing – review & editing:** Xue-Yan Yuan, Ai Cui.

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
