## [Decision Letter · Decision Letter 0]

2 Mar 2020

PONE-D-20-00941

Evaluation of health-related quality of life and the related factors in a group of
Chinese patients with interstitial lung diseases

PLOS ONE

Dear Dr. Ai Cui,

Thank you for submitting your manuscript to PLOS ONE. After careful consideration, we
feel that it has merit but does not fully meet PLOS ONE’s publication criteria as it
currently stands. Therefore, we invite you to submit a revised version of the
manuscript that addresses the points raised during the review process.

Thank
you for your submission to PLOS ONE. Both the reviewers highlighted
the originality of the data and the importance of the topic. However some
fundamental methodological pitfalls need to be addressed.

In particular, the heterogeneity of the population analysed. Given the high number of
diagnoses, with different clinical characteristics and prognosis, that can be
included in the family of interstitial lung diseases, I suggest to restrict the
study to the main diagnoses. Furthermore, the statistical analysis need a complete
revision by an expert statistician.

We would appreciate receiving your revised manuscript by April 14th, 2020. When you
are ready to submit your revision, log on to https://www.editorialmanager.com/pone/ and select the 'Submissions
Needing Revision' folder to locate your manuscript file.

If you would like to make changes to your financial disclosure, please include your
updated statement in your cover letter.

To enhance the reproducibility of your results, we recommend that if applicable you
deposit your laboratory protocols in protocols.io, where a protocol can be assigned
its own identifier (DOI) such that it can be cited independently in the future. For
instructions see: http://journals.plos.org/plosone/s/submission-guidelines#loc-laboratory-protocols

We look forward to receiving your revised manuscript.

Kind regards,

Paola Faverio

Academic Editor

PLOS ONE

Journal Requirements:

**Comments to the Author**

1. Is the manuscript technically sound, and do the data support the conclusions?

Reviewer #1: Partly

Reviewer #2: Yes

2. Has the statistical analysis been performed
appropriately and rigorously? 

Reviewer #1: No

Reviewer #2: Yes

3. Have the authors made all data underlying the
findings in their manuscript fully available?

Reviewer #1: Yes

Reviewer #2: Yes

4. Is the manuscript presented in an intelligible
fashion and written in standard English?

Reviewer #1: Yes

Reviewer #2: Yes

5. Review Comments to the Author

Reviewer #1: General comments

1. The researchers have collected an impressive amount of data, but the study lacks
both structure and focus. The analytical approach is questionable and there are
methodological limitations, in particular with respect to the study population. The
manuscript contains large amounts of information, but adds little new knowledge. It
is not easily accessible to the reader.

2. My main concern is the heterogeneous patient population. The material comprises a
mix of diagnoses including idiopathic interstitial pneumonias (IIP),
hypersensitivity pneumonitis (HP), CTD-ILD and sarcoidosis - all well-defined
diseases with disease-specific clinical characteristics and disease-specific
prognosis. To analyze such a variety of diseases under the same ILD-umbrella is not
meaningful.

3. The study lacks an appropriate purpose. The aim is to “investigate HRQL in
patients with ILD and identify factors influencing HRQL among these patients”. To
what end? What do the researchers want to find out? What is the hypothesis? ILD is a
common denominator for various diseases with different courses, different responses
to therapy and different prognosis. To assess HRQL in a mixed ILD-population is not
useful. I would advise the research group to focus on disease-specific HRQL. Based
on the number of patients in the various disease-groups, it may be an idea to
confine the study to the two largest groups - IIP (69.5%) and CTD-ILD (15%) - and
compare HRQL in those two diseases.

4. The study lacks generalizability. The results of the study may be relevant to the
Chinese medical community, but they cannot be generalized to other parts of the
world. The study was carried out in 2017/18. At that time, anti-fibrotic treatment
was commonly prescribed to patients with IPF in North-America, Europe, Australia and
Japan. In the present study, only 5 patients received such drugs (table 1) while
“other drugs” were used by 74% – presumably drugs that are not generally prescribed
outside of China. Since drugs will affect symptoms, course of disease, side-effects
and prognosis, they will also invariably affect measures of HRQL. This adds to the
difficulties in interpreting and generalizing the results of the study.

Specific comments

Introduction.

Lines 53-64, 65-67 and 67-68. Three statements that all need references.

Methods.

1. The prospective cohort was followed up though “face-to-face or telephone
interviews”. How could “physiological indicators” (like PFTs) be registered by
telephone? What was the procedure for follow-up with respect to the questionnaires?
(lines 84-87)

2. ILDs have insidious onset. How was “date of ILD-diagnosis” defined? (line 96)

3. Equipment, guidelines and ref.values for PFTs should be specified. (lines
98-100)

4. Were the two radiologists blinded to clinical information? (line 101)

5. The minimum clinically important differences in scores should be given for each of
the questionnaires.

Statistics

1. Predictors cannot be identified in a cross-sectional study (lines 152-154)

2. The model seems overloaded (table 3)

3. Multiple correlations are probably not the method of choice for analyzing
between-group changes in the prospective study (lines 157-160).

4. An expert statistician should be consulted with respect to analytical
approach.

Results

1. In general, I suggest dropping the use of two decimal places (text and tables). It
indicates a precision level that is not real. Example: duration of symptoms 19.89
+27.34 months (line 168 + tab1).

2. Tab 2. All variables have large SDs, simply reflecting the fact that wide ranges
will always be found in very heterogeneous study populations. To what end is such
information useful?

3. In the longitudinal study, relatively large proportions of patients had
improvements in both symptoms (dyspnea, cough) and PFTs, and also in HRQ scores.
Patients with ILD tend to deteriorate or remain stable over time, not improve.
Presumably the unexpected findings reflect that in a mixed study population -
including patients with a broad variety of diagnoses/diseases - some will respond to
therapy/drugs while others will not. (lines 246-257).

Reviewer #2: The original research article by Yuan and co-authors assessed
longitudinally the quality of life of 200 ILD patients at baseline, and at 6- and
12-month follow up correlating the changes in HRQoL with the changes in pulmonary
function tests. The authors found that HRQoL deteriorates during the follow up of
ILD patients and seems to be associated with determinants such as dyspnea, cough,
and depression.

Overall the manuscript is well written and comprehensive; therefore I suggest only
minor revisions.

1. Abstract – Introduction – Lines 16-18. The sentence “There are limited therapeutic
options and strong side effects, which eventually lead to respiratory failure and
affect the quality of life of patients” seems to suggest that the treatment-related
side effects cause respiratory failure.

2. Abstract – Results – Line 27. Correct “The mean age was 60.7 years old, …” with
“The mean age was 60.7 years, …”.

3. Introduction – Lines 65-68. I suggest quoting the few available studies dealing
with HRQoL in ILD patients.

4. Results – Line 164. I suggest using the same number of decimals for the median age
and its standard deviation, as it has been done in the whole manuscript for
continuous variables.

5. Results – Table 1. Correct “Ethmic group” with “Ethnic group”.

6. Results – Lines 251-252. Does the sentence “Similarly, 42.4% of patients’ FVC%
predicted and 48.8% of patients’ DLco% predict were improved or stable” refer to the
12-month follow up? If so, it should be specified in the text.

7. Results – Lines 263-264. “The associations between longitudinal changes 263 in
HRQoL and clinical characteristics is shown in Table 4”. I suggest indicating in the
text which clinical characteristics were tested in this analysis.

8. Results – Line 284. Change “the emergence of the first symptom” with “the onset of
the first symptom”.

9. Discussion. This paper is an example of the raising interest in the
patient-reported outcome measures (PROs) in IPF and ILDs in general. However, the
minimal clinically significant difference of the PROs used in this paper (mMRC, LCQ,
HADS, SGRQ and SF-36 questionnaire) is currently unknown. This should be mentioned
among the limitations of this paper.

6. PLOS authors have the option to publish the peer
review history of their article (what does this mean?). If published, this will
include your full peer review and any attached files.

If you choose “no”, your identity will remain anonymous but your review may still be
made public.

**Do you want your identity to be public for this peer review?** For
information about this choice, including consent withdrawal, please see our
Privacy Policy.

Reviewer #1: No

Reviewer #2: No

---

## [Author Response · Author response to Decision Letter 0]

10 Apr 2020

Dear Prof. Paola Faverio, 

Thank you for your kind letter. We revised the manuscript in accordance with your and
the reviewers’ comments and advices, and carefully proof-read the manuscript to
correct spelling, expression and statistical analytical approach errors. Here below
is our description on revision according to your and the reviewers’ comments and
advices. 

Thank you for your submission to PLOS ONE. Both the reviewers highlighted the
originality of the data and the importance of the topic. However some fundamental
methodological pitfalls need to be addressed.

In particular, the heterogeneity of the population analysed. Given the high number of
diagnoses, with different clinical characteristics and prognosis, that can be
included in the family of interstitial lung diseases, I suggest to restrict the
study to the main diagnoses. Furthermore, the statistical analysis need a complete
revision by an expert statistician.

We appreciate all the comments and suggestions from the editors and reviewers, these
will be great of help to improve our present work. We appreciate the suggestion and
totally agreed with your opinion about the heterogeneity of the population. We
restrict the population in our study to idiopathic interstitial pneumonias (IIP) and
CTD-ILD, and then we reanalyzed the data. We also appreciate your reminding about
the statistical analysis. We specifically consulted the expert statistician and
improved the defects in our study.

Reviewer #1: 

General comments

1. The researchers have collected an impressive amount of data, but the study lacks
both structure and focus. The analytical approach is questionable and there are
methodological limitations, in particular with respect to the study population. The
manuscript contains large amounts of information, but adds little new knowledge. It
is not easily accessible to the reader.

Response: Thank you for the comments. We agreed with the reviewer’s opinion on the
defects in data processing and statistical analysis. The data of the study were from
patients with ILDs in china, and there is a large space for study on the quality of
life in these patients in china. In present study, We adjusted the study population
to focus on patients with IIP and CTD-ILD, and we revised the writing of the
manuscript to make it easier for the reader to access the important information
(Line 236-243, Line 282-294, Line 444-452, Table 1, Table 2 ). 

2. My main concern is the heterogeneous patient population. The material comprises a
mix of diagnoses including idiopathic interstitial pneumonias (IIP),
hypersensitivity pneumonitis (HP), CTD-ILD and sarcoidosis - all well-defined
diseases with disease-specific clinical characteristics and disease-specific
prognosis. To analyze such a variety of diseases under the same ILD-umbrella is not
meaningful.

Response: Thank you for the comments. At first, we look forward to exploring a
universal predictive model for ILDs, so we collected all patients clinically
diagnosed with ILDs and followed them up for 12 months. Since ILD belongs to a group
of diseases and the heterogeneity is very significant, it is obviously
inappropriate. We think your views on the heterogeneous patient population are very
reasonable and Based on your suggestion, we have eliminated HP, sarcoidosis and
other types of ILD, and re-analyzed the data (Line 236-521, Table 1, Table 2, Figure
1, Figure 3 ).

3. The study lacks an appropriate purpose. The aim is to “investigate HRQL in
patients with ILD and identify factors influencing HRQL among these patients”. To
what end? What do the researchers want to find out? What is the hypothesis? ILD is a
common denominator for various diseases with different courses, different responses
to therapy and different prognosis. To assess HRQL in a mixed ILD-population is not
useful. I would advise the research group to focus on disease-specific HRQL. Based
on the number of patients in the various disease-groups, it may be an idea to
confine the study to the two largest groups - IIP (69.5%) and CTD-ILD (15%) - and
compare HRQL in those two diseases.

Response: Thanks for your comments and suggestions on our study. We agreed with the
opinion on the aim of the study, and we focused on IIP and CTD-ILD according to your
suggestion. We re-analyzed sociodemographic, disease-related characteristics in both
ILD subtypes and compared HRQL in them. 

4. The study lacks generalizability. The results of the study may be relevant to the
Chinese medical community, but they cannot be generalized to other parts of the
world. The study was carried out in 2017/18. At that time, anti-fibrotic treatment
was commonly prescribed to patients with IPF in North-America, Europe, Australia and
Japan. In the present study, only 5 patients received such drugs (table 1) while
“other drugs” were used by 74% – presumably drugs that are not generally prescribed
outside of China. Since drugs will affect symptoms, course of disease, side-effects
and prognosis, they will also invariably affect measures of HRQL. This adds to the
difficulties in interpreting and generalizing the results of the study.

Response: We appreciate your comment on the generalizability of the research. We are
sorry that our way of expression may has interfered with you. In our study, “others”
included “no intervention”, “antioxidants” and “traditional Chinese medicine”, which
“no intervention” was used in 113 patients (56.2%). We refined the presentation of
data to separate “no intervention” from “others” (table 1). Meanwhile, anti-fibrotic
treatment is not commonly prescribed to patients with IPF due to the price problem,
and anti-fibrotic treatment is only applicable to patients with IPF, not all
patients with ILDs. Theses factors finally leads to the low application of
anti-fibrotic treatment in present study. We agreed with your opinion that drugs may
be the confounding factor affecting HRQL in ILDs. In present study, we focused on
the impact of disease severity on quality of life. Although drugs may have an impact
on quality of life, it is mainly a change in the severity of the diseases after
treatment. We analyzed the effect of drugs factor on HRQL with ILD by linear
regression analysis (table 3), but we are sorry that the statistical analysis cannot
fully explain the real-world problems. 

Specific comments

Introduction.

Lines 53-64, 65-67 and 67-68. Three statements that all need references.

Response: We appreciate for your advice, several publications have been added in
corresponding statements of the article (Line 67-68, Curr Opin Pulm Med. 2013; 19:
474–479, Sarcoidosis Vasc Diffuse Lung Dis. 2016; 33: 341–348, Respirology. 2014;
19: 1019–1924; Line 69-72, Am J Respir Crit Care Med. 2007; 176: 636–643; Line
74-75, Respir Res. 2019; 20: 59–72, Respirology. 2017; 22: 950–956, Respir Res.
2020; 21: 36–47; Line 79-80, Respirology. 2014; 19: 1019–1924, Respiration. 2007;
74: 401–405; Line 81-84, Respir Med. 2017; 127: 1–6, Respirology. 2018, Chest. 2014;
145: 1333–1338, J Bras Pneumol. 2010; 36: 562–570). 

Methods.

1. The prospective cohort was followed up though “face-to-face or telephone
interviews”. How could “physiological indicators” (like PFTs) be registered by
telephone? What was the procedure for follow-up with respect to the questionnaires?
(lines 84-87)

Response: Thank you for your comments, it is indeed our unclear expression in
previous manuscript. During the follow-up, some patients did not complete the
questionnaires for personal reasons after pulmonary function test in our hospital.
We registered the questionnaires by telephone within one week after the completion
of the pulmonary function test. We corrected the inaccurate expression (Line
103-107).

2. ILDs have insidious onset. How was “date of ILD-diagnosis” defined? (line 96)

Response: Thank you for the remind. “Date of ILD-diagnosis” was defined as the time
from symptom onset to the diagnosis of ILD in our hospital. We are sorry for the
ambiguous expression and modified “Date of ILD-diagnosis” to “disease duration”
(Line 116). 

3. Equipment, guidelines and ref.values for PFTs should be specified. (lines
98-100)

Response: Thank you for your advice, we added relevant information in the manuscript
(Line 123-128).

4. Were the two radiologists blinded to clinical information? (line 101)

Response: Yes, we are sorry for our negligence of this information and added it into
the manuscript (Line 129-131). 

5. The minimum clinically important differences in scores should be given for each of
the questionnaires.

Response: Thank you for the remind, the minimum clinically important differences of
PROs using in our manuscript is currently unknow. We mentioned this fact among the
limitations of this paper (Line 778-786).

Statistics 

1. Predictors cannot be identified in a cross-sectional study (lines 152-154)

Response: Thank you for the remind, we modified the presentation of this part (Line
195-196).

2. The model seems overloaded (table 3)

Response: We appreciate for your advice, and we simplified the model and described
the results of univariate and multivariate linear regression analysis separately
(table 3, table 4). 

3. Multiple correlations are probably not the method of choice for analyzing
between-group changes in the prospective study (lines 157-160).

Response: Thank you for the remind, we changed the statistical method from multiple
correlations to linear regression analysis and we further assessed HRQoL using
one-way analysis of variance (ANOVA) followed by pairwise comparisons according to
the Least-Significant-Difference method (Line 213-218).

4. An expert statistician should be consulted with respect to analytical
approach.

Response: Thank you for your advice, we consulted the expert statistician and
corrected the deficiencies (Line 191-195, Line 213-218). In addition, we re-analyzed
the data (Table 1, Table 2, Table 5, S1 Table).

Results

1. In general, I suggest dropping the use of two decimal places (text and tables). It
indicates a precision level that is not real. Example: duration of symptoms 19.89
+27.34 months (line 168 + tab1).

Response: Thank you for your suggestion, we totally agreed that two decimal places
indicates a precision level that is not real. We changed the use of two decimal
places to one decimal place in all text and tables. 

2. Tab 2. All variables have large SDs, simply reflecting the fact that wide ranges
will always be found in very heterogeneous study populations. To what end is such
information useful?

Response: We agreed that variables that have large SDs reflect the very heterogeneous
populations in our study. So, we limited the study population to IIP and CTD-ILD
patients and re-analyzed the data. The results for the two group was added. 

3. In the longitudinal study, relatively large proportions of patients had
improvements in both symptoms (dyspnea, cough) and PFTs, and also in HRQ scores.
Patients with ILD tend to deteriorate or remain stable over time, not improve.
Presumably the unexpected findings reflect that in a mixed study population -
including patients with a broad variety of diagnoses/diseases - some will respond to
therapy/drugs while others will not. (lines 246-257).

Response: Thank you for your comments. We agreed that the heterogeneous study
populations in our study led to the unexpected findings, so we excluded Sarcoidosis,
HP and other ILD subtypes in our analysis. After reanalyzing the data, we found that
symptoms including dyspnea and cough deteriorated or remained stable or in most
patients at follow-up (S1 table). At 6-month and 12-month follow-up, patients with
IIP had a stable quality of life compared with baseline in HRQoL measured by SF-36
PCS, SGRQ total domains. However, the HRQOL in patients with CTD-ILD had a valuable
improvement at 6-month and 12-month follow-up. The reason for the fact is likely to
be the therapy or drugs. We are sorry for our negligence in this factor at follow-up
and we explained this in the discussion (Line 792-793).

Reviewer #2: 

The original research article by Yuan and co-authors assessed longitudinally the
quality of life of 200 ILD patients at baseline, and at 6- and 12-month follow up
correlating the changes in HRQoL with the changes in pulmonary function tests. The
authors found that HRQoL deteriorates during the follow up of ILD patients and seems
to be associated with determinants such as dyspnea, cough, and depression.

Overall the manuscript is well written and comprehensive; therefore I suggest only
minor revisions.

1. Abstract – Introduction – Lines 16-18. The sentence “There are limited therapeutic
options and strong side effects, which eventually lead to respiratory failure and
affect the quality of life of patients” seems to suggest that the treatment-related
side effects cause respiratory failure.

Response: Thank you for the remind, it is indeed our mistake in the expression. The
sentence (Line 16-19) was modified.

2. Abstract – Results – Line 27. Correct “The mean age was 60.7 years old, …” with
“The mean age was 60.7 years, …”.

Response: Thank you for pointing out the error, we have corrected it.

3. Introduction – Lines 65-68. I suggest quoting the few available studies dealing
with HRQoL in ILD patients.

Response: Thank you for your advice, several available studies have been added in
corresponding statements of the manuscript (Line 81-84, Respir Med. 2017; 127: 1–6,
Respirology. 2018. https://doi.org/10.1111/ resp. 13293, Chest. 2014; 145: 1333–1338, J
Bras Pneumol. 2010; 36: 562–570)

4. Results – Line 164. I suggest using the same number of decimals for the median age
and its standard deviation, as it has been done in the whole manuscript for
continuous variables.

Response: Thank you for your advice, we have unified all the median±SD to one number
of decimals in the whole manuscript.

5. Results – Table 1. Correct “Ethmic group” with “Ethnic group”.

Response: Thank you for pointing out the error, we have corrected it. 

6. Results – Lines 251-252. Does the sentence “Similarly, 42.4% of patients’ FVC%
predicted and 48.8% of patients’ DLco% predict were improved or stable” refer to the
12-month follow up? If so, it should be specified in the text.

Response: Yes, thank you for your suggestion. We have made additional explanations in
the text (Line 427-429).

7. Results – Lines 263-264. “The associations between longitudinal changes in HRQoL
and clinical characteristics is shown in Table 4”. I suggest indicating in the text
which clinical characteristics were tested in this analysis.

Response: We appreciate for your advice, and specific clinical characteristics
including FVC% predicted, DLco% predicted, mMRC and LCQ total score have been
described in the manuscript (Line 498-500). 

8. Results – Line 284. Change “the emergence of the first symptom” with “the onset of
the first symptom”.

Response: Thank you for your advice, we have corrected it.

9. Discussion. This paper is an example of the raising interest in the
patient-reported outcome measures (PROs) in IPF and ILDs in general. However, the
minimal clinically significant difference of the PROs used in this paper (mMRC, LCQ,
HADS, SGRQ and SF-36 questionnaire) is currently unknown. This should be mentioned
among the limitations of this paper.

Response: Thank you for your advice, we have elaborated this limitation in the
discussion of the manuscript (Line 778-786).

We acknowledge the editors’ and reviewers’ comments and suggestions very much, which
are

valuable in improving the quality of our manuscript. Thank you again for the guidance
and 

help.

Sincerely yours,

Xue-Yan Yuan, Ai Cui

2020-04-07

to Reviewers.docx
---

## [Decision Letter · Decision Letter 1]

27 May 2020

PONE-D-20-00941R1

Evaluation of health-related quality of life and the related factors in a group of
Chinese patients with interstitial lung diseases

PLOS ONE

Dear Dr. Ai Cui,

Thank you for submitting your manuscript to PLOS ONE. After careful consideration, we
feel that it has merit but does not fully meet PLOS ONE’s publication criteria as it
currently stands. Therefore, we invite you to submit a revised version of the
manuscript that addresses the points raised during the review process.

The quality of the research article have much improved after the revisions made.

I agree with the minor revisions proposed by Reviewer 3.

I also suggest the authors to have the abstract of the manuscript, particularly the
sentences changed during the review, rechecked by an English translator.

If you would like to make changes to your financial disclosure, please include your
updated statement in your cover letter. Guidelines for resubmitting your figure
files are available below the reviewer comments at the end of this letter.

We look forward to receiving your revised manuscript.

Kind regards,

Paola Faverio

Academic Editor

PLOS ONE

Reviewers' comments:

Reviewer's Responses to Questions

**Comments to the Author**

1. If the authors have adequately addressed your comments raised in a previous round
of review and you feel that this manuscript is now acceptable for publication, you
may indicate that here to bypass the “Comments to the Author” section, enter your
conflict of interest statement in the “Confidential to Editor” section, and submit
your "Accept" recommendation.

Reviewer #2: All comments have been addressed

Reviewer #3: All comments have been addressed

2. Is the manuscript technically sound, and do the data
support the conclusions?

Reviewer #2: Yes

Reviewer #3: Partly

3. Has the statistical analysis been performed
appropriately and rigorously? 

Reviewer #2: Yes

Reviewer #3: Yes

4. Have the authors made all data underlying the
findings in their manuscript fully available?

Reviewer #2: Yes

Reviewer #3: Yes

5. Is the manuscript presented in an intelligible
fashion and written in standard English?

Reviewer #2: Yes

Reviewer #3: Yes

6. Review Comments to the Author

Reviewer #2: (No Response)

Reviewer #3: Xue-yan Yuan and collaborators wrote a research article trying to
clarify the role of health-related quality of life factors in a cohort of Chinese
patients affected by ILD.

This topic has been widely discussed in literature, however it is still a subject of
interest. 

Some minor revisions need to be made.

Methods

Line 80. In this section I suggest you to add which ILDs you include in your cohort
(lately in the section Results you state you have considered only ILD patients with
a diagnosis of CTD-ILD or IIP).

Results

Line 180 since idiopathic interstitial pneumonias include different entities as well
as CTD-ILD (Am J Respir Crit Care Med. 2013 Sep 15;188(6):733-48) I suggest you to
specify the composition of your cohort. This could help better understand your
results since in both groups are included entities with different evolution and
prognosis. Moreover it could help understanding why antifibrotic drug were
administered in only 2.9% of the IPP group (were IPF patients the minority of this
group?)

Line 238. “ILD subtype (IIP vs CTD-ILD) was associated with most dimensions of the
SGRQ and the SF-36 PCS” I suggest you to deepen this statement. Are there
significant differences in the results of this questionnaires in IPP patients
compared to CTD-ILD?

In section Results the comparison between IPP and CTD-ILD is clear only in the first
and second paragraphs (patient characteristics and HRQoL) while in the other
paragraphs (factors influencing HRQoL assessed at the time of enrollment;
relationship between the change in HRQoL and changes 274 in clinical characteristics
and clinical characteristics and HRQoL of patients who died during follow-up) the
comparison is absent. In these last paragraphs you consider “ILD subtypes” as a
factor influencing or not HRQoL while I think it could be of interest analyze not
only the whole cohort but also if there are differences between the two groups in
terms of HRQoL.

Moreover, a better focus on differences between the two groups should be highlighted
in the section Discussion and in the section Conclusion.

7. PLOS authors have the option to publish the peer
review history of their article (what does this mean?). If published, this will
include your full peer review and any attached files.

If you choose “no”, your identity will remain anonymous but your review may still be
made public.

**Do you want your identity to be public for this peer review?** For
information about this choice, including consent withdrawal, please see our
Privacy Policy.

Reviewer #2: No

Reviewer #3: Yes: Anna Stainer

---

## [Author Response · Author response to Decision Letter 1]

14 Jun 2020

Dear Prof. Paola Faverio, 

Thank you for your kind letter. We have revised our manuscript after reading the
comments and suggestions provided by you and the reviewers. The following is what we
have modified in response to your and the reviewers’ comments and advice.

I agree with the minor revisions proposed by Reviewer 3.

I also suggest the authors to have the abstract of the manuscript, particularly the
sentences changed during the review, rechecked by an English translator.

We appreciate all your and the reviewers’ comments and advice. According to your
suggestion, we employed an English-language editing service, Academic English
Editors (AJE) to polish our wording of the changed sentences and abstract, and the
revised parts were marked in the red in the “Revised Manuscript with Track Changes”.
We also elaborated on the diagnosis of the included patients in this study.

Responds to the reviewer’s comments: 

Reviewer #3: 

Xue-yan Yuan and collaborators wrote a research article trying to clarify the role of
health-related quality of life factors in a cohort of Chinese patients affected by
ILD.

This topic has been widely discussed in literature, however it is still a subject of
interest. 

Some minor revisions need to be made.

Methods

1. Line 80. In this section I suggest you to add which ILDs you include in your
cohort (lately in the section Results you state you have considered only ILD
patients with a diagnosis of CTD-ILD or IIP).

Response: Thank you for the comment and suggestion. The specific ILDs subtypes have
been described in the manuscript (Line 84, Line 89, and Line 91: “Patients aged ≤85
years who were diagnosed with ILDs including IIP and CTD-ILD at the Department of
Respiratory and Critical Medicine of Capital Medical University”, “The diagnosis of
IIP or CTD-ILD”).

Results

1. Line 180 since idiopathic interstitial pneumonias include different entities as
well as CTD-ILD (Am J Respir Crit Care Med. 2013 Sep 15;188(6):733-48) I suggest you
to specify the composition of your cohort. This could help better understand your
results since in both groups are included entities with different evolution and
prognosis. Moreover it could help understanding why antifibrotic drug were
administered in only 2.9% of the IPP group (were IPF patients the minority of this
group?)

Response: Thank you for the remind. According to your suggestion, we have revised our
manuscript and the specific composition of our included patients has been increased
in the manuscript (Line 200-204: “139 were diagnosed with IIP (101 idiopathic
non-specific interstitial pneumonia (NSIP) cases, 18 unclassifiable IIP cases, 11
respiratory-bronchiolitis-ILD cases, 7 IPF cases, and 2 others), and 30 were
diagnosed with CTD-ILD (12 Sjögren’s syndrome (SS) cases, 6 undifferentiated CTD
cases, 4 polymyositis/dermatomyositis (PM/DM) cases, 3 scleroderma (SSc) cases, 2
rheumatoid arthritis (RA) cases, and 3 others) (S1 Table)”). 

2. Line 238. “ILD subtype (IIP vs CTD-ILD) was associated with most dimensions of the
SGRQ and the SF-36 PCS” I suggest you to deepen this statement. Are there
significant differences in the results of this questionnaires in IIP patients
compared to CTD-ILD?

Response: Yes, patients with IIP had significantly better quality of life than
patients with CTD-ILD in terms of SF-36 PCS, SGRQ activity, SGRQ impact, and SGRQ
total domains in our study which were showed in Table 2. We appreciate your
suggestion, and we have further elaborated the relevant information in the
manuscript (Line 262-267: “As shown in Table 3, HRQoL was found to be significantly
affected by multiple factors. ILD subtype was negatively associated with the SF-36
PCS and positively associated with most dimensions of the SGRQ in the univariate
liner regression analyses. As previously mentioned, patients with CTD-ILD had a
lower quality of life as measured by the SF-36 PCS and SGRQ activity, impact, and
total domains when compared with patients with IIP”).

3. In section Results the comparison between IIP and CTD-ILD is clear only in the
first and second paragraphs (patient characteristics and HRQoL) while in the other
paragraphs (factors influencing HRQoL assessed at the time of enrollment;
relationship between the change in HRQoL and changes in clinical characteristics and
clinical characteristics and HRQoL of patients who died during follow-up) the
comparison is absent. In these last paragraphs you consider “ILD subtypes” as a
factor influencing or not HRQoL while I think it could be of interest analyze not
only the whole cohort but also if there are differences between the two groups in
terms of HRQoL.

Response: Thanks for your comments and suggestions. In this study, we mainly focused
on the quality of life of all patients with ILDs and their influencing factors.
Meanwhile, ILD subtypes were weakly associated with SF-36 PCS. So, we did not
analyze the influencing factors in HRQoL of IIP and CTD-ILD patients, respectively.
We compared the quality of life of IIP and CTD-ILD patients, and found HRQoL of
patients with IIP was better than that of patients with CTD-ILD. 

4. Moreover, a better focus on differences between the two groups should be
highlighted in the section Discussion and in the section Conclusion.

Response: We appreciate for your advice and agree with the opinion. We had an
in-depth discussion of the differences between the two groups. (Line 376-378, Line
402-409, Line 415, Line 419-424, Line 504-506)

Finally, we acknowledge the editor’ and reviewers’ comments and suggestions very
much, which are valuable in improving the quality of our manuscript. Thank you again
for the guidance and help.

Sincerely yours,

Xue-Yan Yuan, Ai Cui

2020-06-14

to Reviewers.docx
---

## [Decision Letter · Decision Letter 2]

7 Jul 2020

Evaluation of health-related quality of life and the related factors in a group of
Chinese patients with interstitial lung diseases

PONE-D-20-00941R2

Dear Dr. Ai Cui,

We’re pleased to inform you that your manuscript has been judged scientifically
suitable for publication and will be formally accepted for publication once it meets
all outstanding technical requirements.

Kind regards,

Paola Faverio

Academic Editor

PLOS ONE

Additional Editor Comments (optional):

The issues reasonably raised by the reviewers have been addressed.

Reviewers' comments:

Reviewer's Responses to Questions

**Comments to the Author**

1. If the authors have adequately addressed your comments raised in a previous round
of review and you feel that this manuscript is now acceptable for publication, you
may indicate that here to bypass the “Comments to the Author” section, enter your
conflict of interest statement in the “Confidential to Editor” section, and submit
your "Accept" recommendation.

Reviewer #3: All comments have been addressed

2. Is the manuscript technically sound, and do the data
support the conclusions?

Reviewer #3: Yes

3. Has the statistical analysis been performed
appropriately and rigorously? 

Reviewer #3: Yes

4. Have the authors made all data underlying the
findings in their manuscript fully available?

Reviewer #3: Yes

5. Is the manuscript presented in an intelligible
fashion and written in standard English?

Reviewer #3: Yes

6. Review Comments to the Author

Reviewer #3: Xue-yan Yuan and collaborators wrote a research article trying to
clarify the role of

health-related quality of life factors in a cohort of Chinese patients affected by
ILD.

This topic has been widely discussed in literature, however it is still a subject
of

interest.

All requested corrections were made by the Authors. No further revisions are
required

7. PLOS authors have the option to publish the peer
review history of their article (what does this mean?). If published, this will
include your full peer review and any attached files.

If you choose “no”, your identity will remain anonymous but your review may still be
made public.

**Do you want your identity to be public for this peer review?** For
information about this choice, including consent withdrawal, please see our
Privacy Policy.

Reviewer #3: **Yes: **Anna Stainer

---

## [Editor Report · Acceptance letter]

17 Jul 2020

PONE-D-20-00941R2 

Evaluation of health-related quality of life and the related factors in a group of
Chinese patients with interstitial lung diseases 

Dear Dr. Cui:

I'm pleased to inform you that your manuscript has been deemed suitable for
publication in PLOS ONE. Congratulations! Your manuscript is now with our production
department. 

Kind regards, 

on behalf of

Dr. Paola Faverio 

Academic Editor

PLOS ONE